# Optimized Tokenization for Transcribed Error Correction

**Tomer Wullach**
OriginAI
tomerw@originai.co

**Shlomo E. Chazan**
OriginAI
shlomi@originai.co

## Abstract

The challenges facing speech recognition systems, such as variations in pronunciations, adverse audio conditions, and the scarcity of labeled data, emphasize the necessity for a post-processing step that corrects recurring errors. Previous research has shown the advantages of employing dedicated error correction models, yet training such models requires large amounts of labeled data which is not easily obtained. To overcome this limitation, synthetic transcribed-like data is often utilized, however, bridging the distribution gap between transcribed errors and synthetic noise is not trivial. In this paper, we demonstrate that the performance of correction models can be significantly increased by training solely using synthetic data. Specifically, we empirically show that: (1) synthetic data generated using the error distribution derived from a set of transcribed data outperforms the common approach of applying random perturbations; (2) applying language-specific adjustments to the vocabulary of a BPE tokenizer strike a balance between adapting to unseen distributions and retaining knowledge of transcribed errors. We showcase the benefits of these key observations, and evaluate our approach using multiple languages, speech recognition systems and prominent speech recognition datasets.

## 1 Introduction

Speech Recognition systems has undergone remarkable progress in the last several years, largely due to the growing adoption of the self-supervision paradigm. (Baevski et al., 2020; Hsu et al., 2021; Radford et al., 2022; Balestriero et al., 2023). These systems can attain low error rates given sufficient model capacity and quality labeled data, yet they are susceptible to errors caused by audio and speech factors, such as speakers pronunciation, background noise, recording quality and orthographic errors which occurs mostly in non-phonetic languages.

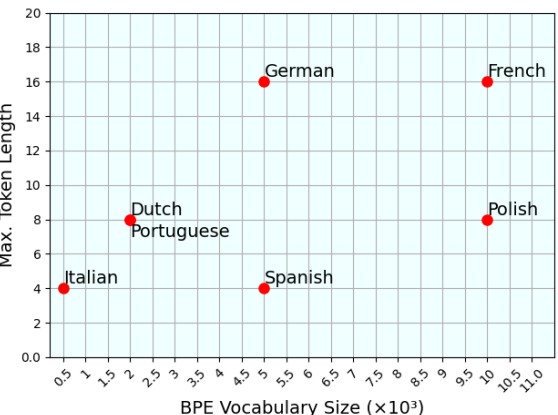

Figure 1: The best performing BPE hyperparameters for seven European correction models. Phonetic (isolating, low morpheme-word ratio) languages tend to use short tokens with small vocabularies. Morphological-rich (bundles several morphemes in each word, inflectional patterns) requires longer tokens and larger vocabularies to memorize inflected words and complex structures.

To this end, researchers suggested text-to-text error correction methods specifically designed to correct transcribed errors (Mani et al., 2020; Leng et al., 2021a; Hrinchuk et al., 2020). These transformer-based methods typically requires a massive and diverse training data (i.e., transcribed audio segments and their corresponding human-authored transcriptions) which is rarely at reach.

To bridge that gap, synthetic data is frequently incorporated to ensure a sufficient amount of training examples. A common approach is to introduce random perturbations or use predefined confusion sets (Ji et al., 2021; Karpukhin et al., 2019; Belinkov and Bisk, 2017). Nevertheless, this approach might result with perturbations that are unrelated to transcribed errors, thereby failing to accurately replicate the distribution of such errors.

A promising solution would involve generating a set of synthetic examples that follows the error distribution of transcribed data, and utilize it to

train a correction model. Such model needs to find a balance between memorizing the transcribe-like patterns by fitting the synthetic training data, and generalizing to unseen distributions.

Transformer-based correction models mostly use subword tokenizers such as BPE (Sennrich et al., 2015), which merges tokens based on a learned set of rules proportional to the tokenizer's vocabulary size. Prior research showed that small vocabularies consisting of short tokens, such as individual characters, may assist with spelling errors, natural noise and generalize to unseen distributions (Ma et al., 2020; Kann et al., 2020). However, it may also lead to degraded performance in some settings (Gowda and May, 2020; Clark et al., 2022).

Alternatively, it has been shown that large vocabularies can affect the model's memorization capabilities (Kharitonov et al., 2021), which can be significant for recurring transcribed error patterns.

To address these shortcomings, we generate examples that mimic the error distribution of transcribed texts by incorporating information derived from transcribed errors. We then balance between memorizing erroneous patterns and generalizing to unseen distributions using language-specific adjustments to the vocabulary of BPE tokenizers. These adjustments limit the vocabulary size and the maximum length of vocabulary tokens.

Interestingly, we find that the optimal vocabulary limitations differs between languages (Figure 1). Our experiments reveal a correlation between the vocabulary size and the phonetic complexity of the languages. Phonetically simpler languages tend to perform better with smaller BPE vocabularies. Furthermore, we have noticed that languages with rich morphological structures benefit from a longer token length limitation. One possible explanation is that non-phonetic languages with extensive morphology structures require memorizing error patterns that consist of longer sequences. The field of research concerning the maximum length of individual vocabulary tokens, irrespective of vocabulary size, is currently lacking in extensive study.

Our contributions can be summarized as follows:

1. We demonstrate the advantages of training error correction models with synthetic data derived from transcribed errors and an adjusted BPE vocabulary.

2. Our approach generalizes across diverse datasets and outputs from various speech recognition models, despite being trained exclusively using synthetic data.

3. We provide thorough experiments and ablation, demonstrating the trade-offs in adjusting language-specific BPE vocabulary sizes.

## 2 Related Work

**Transcribed Errors Correction.** Correction methods are commonly applied as a post-processing step, aiming to map transcribed errors to their correct form. Recent works demonstrated the merits of utilizing transformer-based architectures, stretching from auto-regressive (Mani et al., 2020; Shen et al., 2022) to non auto-regressive (Leng et al., 2021b,a) approaches.

One of the concerns in using these approaches is the alignment problem, which includes determining the correspondence between source tokens and target tokens. Various methods have been explored to address this issue. For example, some approaches utilize a token duration predictor (Leng et al., 2021b,a), while others leverage the speech recognition system's Connectionist Temporal Classification (CTC) (Graves et al., 2006) predictions (Leng et al., 2022). Moreover, some studies focus on detecting transcribed errors, allowing the correction model to copy tokens identified as correct (Leng et al., 2022; Gekhman et al., 2022).

However, these methods require a significant amount of training data, consisting of pairs of transcribed segments and their corresponding ground-truth transcriptions. Unfortunately, publicly available resources are often insufficient, and obtaining labeled data can be expensive and time-consuming.

To overcome this challenge, a commonly used approach involves generating synthetic training examples. These examples are intentionally noised to resemble transcribed sequences and are then combined with the labeled examples for training the correction model. The perturbations can be introduced through random noise (Shen et al., 2022; Leng et al., 2021b; Ji et al., 2021), predefined confusion sets like homophone dictionaries (Shen et al., 2022), or by using predictions from a pre-trained model (Leng et al., 2022).

We hypothesize that simply incorporating random perturbations or aiming for a similar overall error rate as a labeled example set is insufficient. Instead, it is crucial to closely align with the distribution of transcribed errors. Throughout this

study, we provide empirical evidence to support the validity of this hypothesis.

**Robustness to Perturbations.** Speech recognition systems commonly include a language model in their post-processing stage. The likelihood score produced by the language model allows prioritizing multiple transcribed candidates (Baevski et al., 2020; Hsu et al., 2021; Babu et al., 2021; Chen et al., 2022), and act as a correction mechanism to potentially corrupt transcriptions. However, the transcribed examples may contain character-level errors such as insertions, deletions, and substitutions, which can undermine the effectiveness of the language model (Moradi and Samwald, 2021; Sun et al., 2020; Belinkov and Bisk, 2017).

Previous studies have demonstrated the benefits of employing small vocabularies, such as character-level representations, when dealing with noisy or misspelled texts (Ma et al., 2020; Belinkov and Bisk, 2017; Boukkouri et al., 2020). Inspired by this line of research, we conduct experiments with low-granularity representations to correct errors.

Our research is also related to a body of work that focuses on creating robust representations through the regularization of the tokenization process (Provilkov et al., 2019; Kudo, 2018). Our work stands out in that our objective is to discover an optimal segmentation that is specifically tailored for transcribed errors in a particular language.

## 3 Methodology

In this work, we create transcribed errors correction models that acts as a post-processing step, correcting the outputs of a speech recognition model.

We derive the error distribution from transcribed labeled data and leverage it to create synthetic examples. To capture an informative error distribution rather than random one, we transcribe audio segments using fine-tuned XLS-R models and compare the transcribed outputs with ground-truth transcriptions. Based on the extracted distribution, we introduce perturbations to a large set of text sequence, resulting in synthetic examples that resemble transcribed data. These synthetic examples are subsequently utilized to train our error correction models, as well as BPE tokenizers.

We aim to control the model's ability to generalize to unseen distributions and memorize transcribed error patterns by adjusting the BPE vocabulary. Concretely, we experiment with the vocabulary size and the maximum length of each token.

### 3.1 Simulating Transcribed Data.

Let $G$ be a model trained to transcribe audio segments of language $X$, and $D$ be a set of labeled examples, i.e., audio segments and their corresponding transcriptions. We transcribe samples of $D$ using $G$ and extract the error distribution. We then generate synthetic examples by applying the error distribution extracted by using $G$ to transcribe $D$.

The transcribed error distribution is extracted by quantifying the operations required to transform a sentence $d_i \in D$ into $\tilde{d}i$, its transcribed form predicted using $G$. Specifically, we define $del(\tilde{d}i, d_i, c)$ as the number of times a character $c \in d_i$ was deleted while transforming to $\tilde{d}i$, $insert(\tilde{d}i, d_i, c, x, y)$ as the number of times $c$ was inserted between characters $x, y \in d_i$, and $subs(\tilde{d}i, d_i, c, x)$ as the number of times a character $c \in d_i$ was replaced with a character $x \in \tilde{d}i$.

We then compute the error probability for each character $c \in D$ by summing over each error type, normalized by the frequency of $c'$ in $D$ ($|c|$). The probability for deleting, inserting and substituting each character is detailed in equations 1a-1c.

$$P_{del}(c) = \frac{\sum\limits_{d \in D} del(\tilde{d}_i, d_i, c) + \gamma}{|c|}, \quad (1a)$$

$$P_{insert}(c, x, y) = \frac{\sum\limits_{d \in D} insert(\tilde{d}_i, d_i, c, x, y)}{|c|}, \quad (1b)$$

$$P_{subs}(c, x) = \frac{\sum\limits_{d \in D} subs(\tilde{d}_i, d_i, c, x) + \gamma}{|c|}, \quad (1c)$$

We observed that performance can be improved by including a smoothing factor $\gamma = \alpha \cdot |c|$, where $\alpha$ is a hyperparameter and $|c|$ is the frequency of the character $c$ in the dataset $D$. We posit that it can be attributed to the underrepresentation of specific errors when comparing different datasets.

Next, we utilize a raw corpora $D_r$ to generate synthetic examples. To apply the error distribution to a raw sentence $d_r \in D_r$, we split the sentence into words based on spaces, and then iterate over each character $c \in d_r$ and delete it with $P_{del}(c)$ probability or substitute it with character $x$ with $P_{subs(c,x)}$ probability. Then, we take each two consecutive characters $x, y \in d_r$ and insert a character $c$ between them with probability $P_{insert}(c, x, y)$.

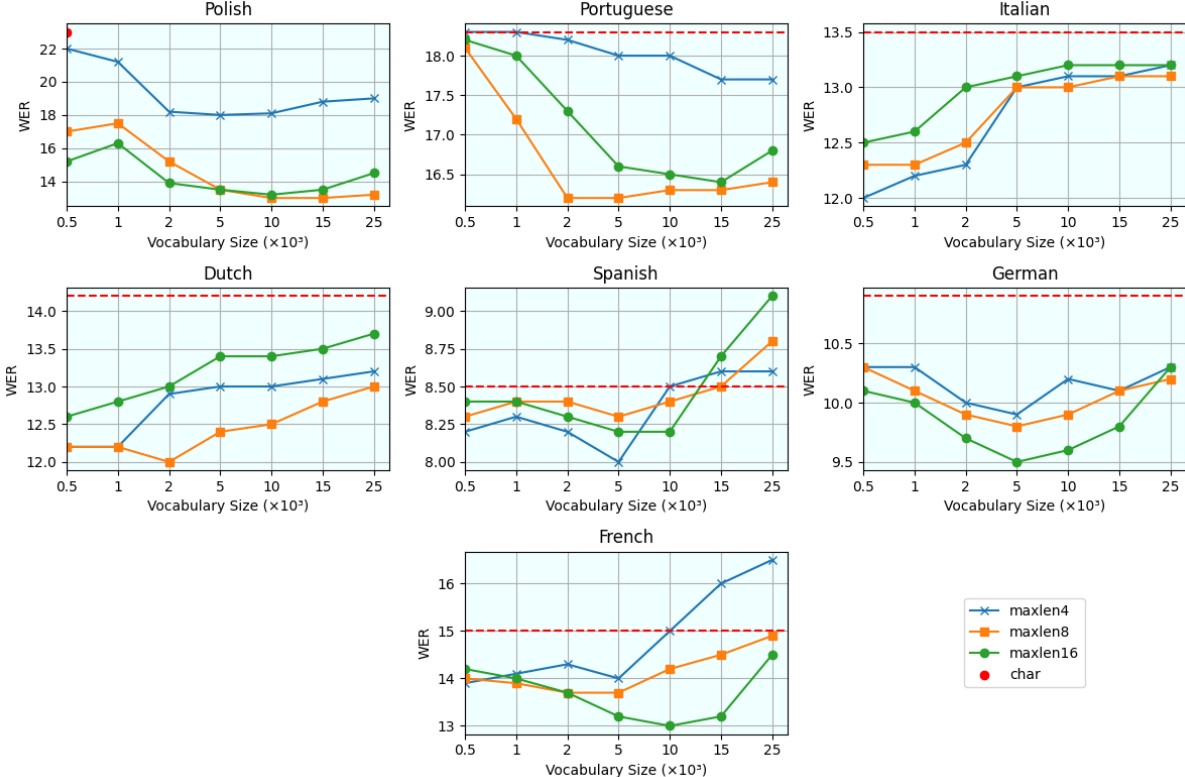

Figure 2: Error correction performance (WER) on MLS (Pratap et al., 2020) validation sets using varying levels of vocabulary size and maximum token length limitations. The results were obtained by employing XLS-R models fine-tuned for each specific language. The "char" horizontal lines indicates the performance using a vocabulary consisting of individual characters found in the training data.

To prevent excessively noisy synthetic examples, we restrict the number of changes made to each word in $d_r$ to a maximum of one. We utilize these synthetic examples to train an error correction model that converts the noisy and corrupted sentences into their correct counterparts.

### 3.2 BPE Vocabulary Adjustments.

BPE vocabulary size has a significant impact on the processing of input sequences, as it directly affects the number of segmented tokens (Gowda and May, 2020). Using a smaller vocabulary leads to a coarser segmentation, which can be beneficial for generalizing to unseen distributions and handling transcribed errors effectively.

On the other hand, a larger vocabulary enhances the model's ability to memorize patterns and less frequent words. Prior studies have shown that vocabulary size is correlated with the memorization of the training data (Kharitonov et al., 2021).

We make a significant observation that languages with rich morphology, where words are often composed of multiple morphemes and inflec-

tions, require additional support for memorizing non-phonetic phrases. To address this challenge, we conducted experiments by providing segmented tokens with additional context in the form of maximum token length.

Therefore, our goal is to strike a balance between a smaller vocabulary that promotes generalization and a larger vocabulary that facilitates memorization. For each language included in our experiments, we explore different vocabulary hyperparameters, such as vocabulary size and maximal token length, to determine the best performance for a correction model.

## 4 Experimental Setup

**Transcribed Error Correction.** We conduct experiments with seven European languages and utilize correction models trained on synthetically generated examples. Our experiments includes correcting texts transcribed using several speech recognition models. We employ BPE tokenizers with vocabulary sizes ranging from 500 to 25K tokens, and a token length limitation varying from 4 to

16 characters. For a coarse-grained segmentation comparison, we utilize a character-based tokenizer consisting of the characters found in the training sets. Additionally, we compare our approach to randomly generated synthetic noise at the character level. We follow the procedure described in Leng et al. (2021b) to create randomly noised sentences along with their corresponding ground truth counterparts.

**Generalizing to Unseen Distributions.** We conduct two experimental settings to evaluate the generalization abilities of our approach. Firstly, we perform a cross-dataset evaluation to test the performance on datasets not included in the training process. Secondly, we evaluate our approach on samples transcribed using a different speech recognition model, Whisper (Radford et al., 2022), to assess its ability to capture unique patterns specific to speech recognition. This evaluation is particularly important since our approach is trained using errors obtained by XLS-R models, which have different training objectives than Whisper and may yield different error distributions.

## 4.1 Models.

**Speech Recognition Models.** We utilize pre-trained Whisper (Radford et al., 2022) and XLS-R (Babu et al., 2021) models from the HuggingFace platform[1], with varying number of model parameters, ranging from 39M to 1B.

XLS-R is a speech processing model composed of feature extracting convolution layers followed by multiple transformer encoder layers. It was pre-trained using a contrastive objective to map masked audio spans to latent representations. The model was then fine-tuned using Connectionist Temporal Classification (CTC) (Graves et al., 2006) loss to project contextualized speech representations to the vocabulary space. We utilize a 4-gram language model when reporting the XLS-R models results.

Whisper, a transformer encoder-decoder model, has gained renown for its exceptional speech recognition performance. It was trained using weak supervision on diverse tasks and languages. A key advantage lies in its ability to achieve outstanding results without fine-tuning for specific languages or domains, relying solely on its pre-training.

**Error Correction Model.** We employ a standard transformer encoder-decoder (Vaswani et al., 2017)

architecture, comprised of 6 encoder and decoder layers. We use 8 attention heads in each layer, and the dimension of the feed-forward layers and the embedding layers are set to 2048 and 512, respectively. The input sequences, i.e., the transcribed texts produced by the speech recognition models, are tokenized using a BPE tokenizer.

## 4.2 Data.

Our experiments covers seven languages: Polish (pl), Portuguese (pt), French (fr), German (de), Dutch (nl), Italian (it) and Spanish (es). This blend allows us to explore languages with both phonetic and non-phonetic charasteristics, as well as different degrees of morphological complexity.

**Multilingual Librispeech** Multilingual Librispeech (MLS) (Pratap et al., 2020) consists of a total of 50K hours of read audiobooks in eight european languages. We follow Pratap et al. (2020) and segment the MLS dataset into training, development and test partitions.

We further divide the training partition into two parts: 90% and 10%. The 90% portion is used for fine-tuning XLS-R models, while the remaining 10% is utilized to extract the transcribed error distribution. Except for the Polish language, where we fine-tune the XLS-R models using approximately 98 hours, we utilize the remaining training data to extract the error distribution.

**CommonVoice** CommonVoice (Ardila et al., 2019), a crowed-sourced collection of multilingual human voices, collected through volunteer contributions. We employ CommonVoice to evaluate the ability of our approach to adapt to diverse distributions that were not encountered during training.

**mC4.** We generate synthetic transcribed texts by utilizing the mC4 (Xue et al., 2020) data set, a variant of the Common Crawl dataset that consists of 101 languages. For each language, we randomly select a set of 20M sentences that ranges from 3 to 15 words long. We use lowercase synthetic texts, remove punctuation marks and eliminate digits.

**Synthetic Data Generation.** We utilize fine-tuned XLS-R models to transcribe a dedicated set of examples for extracting error distributions. We then calculate the probabilities of character insertions, deletions, and substitutions, as described in Section 3. Subsequently, we generate pairs of synthetic sentences that resemble transcriptions, ac-

---

[1]https://huggingface.co/

| Model | | de | nl | es | it | pt | pl | fr |
|---|---|---|---|---|---|---|---|---|
| XLS-R (0.3B) | Baseline | 9.0 | 13.5 | 8.1 | 13.1 | 17.0 | 13.9 | 12.4 |
| | Correction | 9.2 | 11.8 | 7.5 | 12.4 | 16.1 | 12.8 | 12.4 |
| XLS-R (1B) | Baseline | 7.4 | 11.6 | 7.1 | 12.0 | 15.8 | 10.5 | 10.2 |
| | Correction | 7.4 | 10.5 | 5.6 | 10.7 | 13.2 | 9.3 | 9.8 |
| Whisper tiny | Baseline | 24.9 | 39.4 | 19.2 | 41.7 | 31.3 | 34.2 | 36.8 |
| | Correction | 24.2 | 36.6 | 16.8 | 37.5 | 28.7 | 33.7 | 36.4 |
| Whisper base | Baseline | 17.7 | 28.4 | 12.8 | 31.1 | 21.9 | 22.8 | 26.6 |
| | Correction | 16.9 | 26.1 | 10.3 | 27.6 | 18.8 | 22.4 | 25.9 |
| Whisper small | Baseline | 10.5 | 17.2 | 7.8 | 21.4 | 13.0 | 11.2 | 16.2 |
| | Correction | 9.8 | 15.7 | 6.6 | 18.4 | 12.2 | 11.0 | 15.8 |
| Whisper medium | Baseline | 7.4 | 11.7 | 5.3 | 16.0 | 9.0 | 6.5 | 8.9 |
| | Correction | 7.1 | 10.5 | 5.3 | 15.3 | 8.8 | 6.5 | 8.9 |
| Whisper large | Baseline | 6.6 | 10.2 | 5.4 | 14.3 | 9.2 | 6.6 | 8.9 |
| | Correction | 6.7 | 10.0 | 5.4 | 14.0 | 8.4 | 6.1 | 8.5 |

Table 1: Speech recognition performance (WER) on MLS test sets. The results of the XLS-R models were obtained using a 4-gram language model.

companied by their unmodified counterparts serving as the ground truth. These parallel sentences are then used to train our correction models.

Furthermore, we explore the impact of the amount of labeled data on the quality of synthetic data. We train correction models using synthetic examples that were generated based on the errors extracted from varying amounts of labeled data.

### 4.3 Training Details.

We follow the training settings of Babu et al. (2021) and fine-tune the XLS-R models 20K updates. We employ the Adam optimizer with 10% warm-up updates, followed by a constant learning rate for the next 40% of updates, which gradually decays to zero in the remaining steps. As for the Whisper model, we do not apply any additional training and report the results solely based on the predictions provided by the pre-trained models.

The correction models are trained using 4 NVDIA V100 GPUs for 300K updates. We use a batch size of 256 samples per device, and accumulate the gradients for every 4 updates. The model's parameters are optimized using the Adam optimizer with a learning rate of 3e-5 and use the first 1000 updates as warm-up. We set a dropout probability to 0.1. We use SentencePiece[2] (Kudo and Richardson, 2018) to train our BPE tokenizers. The tokenizers were trained using the generated synthetic data, while controling the vocabulary size and the maximum token length parameters.

[2]https://github.com/google/sentencepiece

## 5 Results

### 5.1 Transcribed Errors Correction.

The results of our approach on the MLS and CommonVoice datasets are reported in Table 1 and Table 2, respectively. We observe that our approach is particularly effective for the smaller models (e.g., Whisper tiny, base and small). The results demonstrate consistent improvement across multiple languages and showcase the strong generalization performance of our approach on unseen distributions, including different datasets such as CommonVoice and speech recognition models like Whisper.

We also observe significant perplexity reductions when comparing our correction models to a 4-gram language models utilized for post-processing. This provides further motivation regarding the advantages of correction models. That is, they make an auto-regressive prediction for each source token, while a language model maximizes the sentence-level likelihood score which could deviate from the source sequence and cause hallucinations.

Table 3 shows a comparison of different methods for handling perturbations. Our suggested approach outperforms correction models with similar architectures but different training methodologies. Specifically, we trained the models on random perturbations, as well as models trained using the popular method for learning robust representations, BPE dropout (Provilkov et al., 2019).

| Model | | de | nl | es | it | pt | pl | fr |
|---|---|---|---|---|---|---|---|---|
| XLS-R (0.3B) | Baseline | 8.8 | 12.3 | 8.6 | 13.7 | 11.0 | 14.8 | 24.5 |
| | Correction | 8.3 | 11.5 | 8.2 | 12.9 | 10.9 | 14.9 | 23.2 |
| XLS-R (1B) | Baseline | 8.5 | 11.5 | 8.1 | 13.2 | 9.4 | 12.7 | 18.2 |
| | Correction | 8.5 | 11.2 | 7.7 | 12.7 | 8.8 | 12.6 | 17.1 |
| Whisper tiny | Baseline | 34.5 | 43.6 | 30.3 | 44.5 | 35.2 | 45.3 | 49.7 |
| | Correction | 31.3 | 37.1 | 27.2 | 40.3 | 31.3 | 43.1 | 45.4 |
| Whisper base | Baseline | 24.5 | 29.5 | 19.6 | 30.5 | 23.7 | 32.8 | 37.3 |
| | Correction | 22.8 | 25.1 | 17.0 | 27.4 | 22.1 | 30.3 | 32.3 |
| Whisper small | Baseline | 13.0 | 14.2 | 10.3 | 16.0 | 12.5 | 16.9 | 22.7 |
| | Correction | 12.1 | 13.3 | 8.1 | 13.0 | 12.0 | 15.8 | 21.5 |
| Whisper medium | Baseline | 8.5 | 8.0 | 6.9 | 9.4 | 8.1 | 10.1 | 16.0 |
| | Correction | 8.3 | 7.4 | 6.7 | 8.8 | 8.1 | 10.0 | 15.7 |
| Whisper large | Baseline | 7.7 | 7.1 | 6.4 | 8.1 | 7.1 | 9.0 | 14.7 |
| | Correction | 7.9 | 7.0 | 6.0 | 7.8 | 6.9 | 8.6 | 14.7 |

Table 2: Speech recognition performance (WER) on CommonVoice Speech test sets. The results of the XLS-R models were obtained using a 4-gram language model.

| Method\Model | de | nl | es | it | pt | pl | fr |
|---|---|---|---|---|---|---|---|
| Whisper small | 10.5 | 17.2 | 7.8 | 21.4 | 13.0 | 11.2 | 16.2 |
| Random noise (Leng et al., 2021b) | 10.9 | 18.1 | 7.2 | 18.3 | 13.0 | 11.5 | 16.2 |
| BPE dropout (Provilkov et al., 2019) | 11.1 | 15.5 | 7.9 | 22.2 | 12.5 | 11.8 | 16.8 |
| Extracted error dist. + Adjusted BPE (this work) | 9.8 | 15.7 | 6.6 | 18.4 | 12.2 | 11.0 | 15.8 |

Table 3: The performance (WER) of different methods for handling transcribed errors. The results were obtained by employing transformer-based correction models, as well as Whisper small on the CommonVoice dataset.

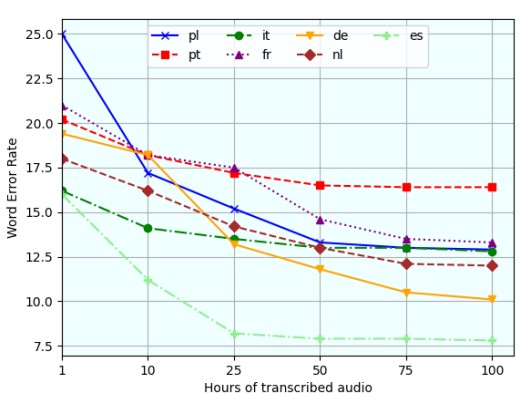

Figure 3: The performance (WER) of correction models using information extracted from varying amounts of transcribed examples. Languages which are considered phonetic (es, it, pt) achieve good performance with as little as 25 hours of labeled data. However, languages that are less phonetic (pl, de, nl) require significantly more data to perform well.

## 5.2 The Effect of Vocabulary Limitations.

A key observation is that the best performing vocabulary size and maximal token differs between

languages, as depicted in Figure 2. For instance, an Italian correction model favors shorter tokens and requires a vocabulary of 500 tokens, while a Polish correction model which benefits from longer tokens and a larger vocabulary. In General, the results indicates that the vocabulary limitations are shared between languages with similar attributes. Figure 1 shows that phonetic languages tend to benefit from small vocabularies and short tokens, while morphological rich languages typically requires larger vocabularies and longer tokens. A possible explanation is that the words of phonetic (isolating) languages are widely composed of one to very few morphemes.

## 5.3 Does Additional Labeled Data Affects The Distribution of Errors?

Figure 3 shows the effect of information extracted from varying amounts of labeled data on correction models. We transcribed the audio samples using 1B parameters XLS-R models, fine-tuned using MLS for each of the languages.

The results indicates that correction models of phonetic languages (*it*, *es*, pt) are able to perform

well with as few as 25 hours of transcribed audio, while languages considered as less phonetic (*de*, *pl*, fr, nl) requires larger amounts in order to show increased performance.

We also observe that the amount of pre-training data is not necessarily correlated with the effectiveness of the extracted error distribution. That is, we expected that languages with higher presence during pre-training would require less data in order to yield a quality transcribed error distribution, yet it is the phonetic attributes which has the most influence (e.g., pre-training consists of ~25K hours of fr data and ~17K hours of pt, yet pt correction model performs well with only ~25 hours for extracting the error distribution, while fr requires much more).

## 6 Conclusions.

In this paper, we propose a transcribed errors correction models trained on synthetic data generated using the errors derived from transcribed examples. We introduce a noise protocol that leverages the error distribution extracted from transcribed samples, and demonstrate the importance of adjusting the vocabulary size and the maximum token length of the models' BPE tokenizer. Moreover, we show that such vocabulary adjustments varies between languages, as they are correlated with their levels of phonetic and morphological properties.

We highlight the concerns and emphasize the limitations associated with training error correction models using synthetic data generated by applying random noise. We showcase the generalization abilities and the performance gains of our proposed approach using several datasets and speech recognition models. The claims made in the paper are supported by a comprehensive experimental study. We hope that this research will inspire further exploration of language-specific adjustments in various natural language processing tasks and domains.

## Limitations

Our proposed method utilizes error distributions derived from transcribed data and employs a language-specific tokenizer to segment the data. Our experiments primarily focused on European languages, specifically those belonging to the Romance, Slavic, and Germanic language families. However, we have not thoroughly validated our approach on languages from other language families, including those commonly spoken in Asian,

African, and Middle Eastern regions.

Additionally, our experiments involved using a fixed set of BPE vocabulary size levels for each language we investigated. Nevertheless, further improvements could be achieved by conducting experiments with additional levels of vocabulary sizes. To generate synthetic examples, we utilized the mC4 dataset. However, it would be beneficial to explore matching the raw dataset to the specific domain of the speech data in future studies.

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
