# OpenReview forum: "Optimized Tokenization for Transcribed Error Correction"
_EMNLP/2023/Conference — EMNLP 2023 Main_

### Official Review · Reviewer_z7J7 · 2023-08-04

**Soundness:** 4

**Excitement:**

4: Strong: This paper deepens the understanding of some phenomenon or lowers the barriers to an existing research direction.

**Missing References:**

References are OK


**Paper Topic And Main Contributions:**

This paper describes an approach to train an error correction model, that is then used to correct the transcripts generated by an ASR system. The key contribution of the paper is that the error correction model is trained on synthetically generated text data, using a process that mimics ASR errors. A synthetic text dataset is generated by applying an error-distribution model estimated from ASR transcripts generated by the XLS-R model, on the mC4 dataset. The synthetic dataset is used to train an error-correction model, and the study highlights that the performance of the error correction model can be optimized by controlling the vocabulary size of the tokenizer, and the max token length. Word Error Rates are reported on the multi-lingual LibriSpeech and CommonVoice datasets. The proposed error correction approach outperforms an LM-based post-processing for error correction.


**Questions For The Authors:**

1. Why is insertion contextual, while deletions/substitutions are not?
1. Any reason to only apply the smoothing factor to del and sub?


**Reasons To Accept:**

* The results are convincing as the approach leads to WER reduction by applying error-correction on the output of a state-of-the-art ASR system (Whisper).
* The work is evaluated on multiple languages, and illustrates the language-dependent nature of the optimal tokenizer vocabulary size and max token length.


**Reasons To Reject:**

Nothing I can think of. Good paper overall.

**Reproducibility:**

4: Could mostly reproduce the results, but there may be some variation because of sample variance or minor variations in their interpretation of the protocol or method.

**Reviewer Confidence:**

5: Positive that my evaluation is correct. I read the paper very carefully and I am very familiar with related work.

**Typos Grammar Style And Presentation Improvements:**

Line 490, did you mean 25 hours? (instead of 25k hours)

---

> ### Author Rebuttal · Authors · 2023-08-28
>
> Thank you for your feedback and comments.
>
> Regarding the first question, although it does make sense that all error types should be contextualized, we empirically found that only an inserted character (space included) depends on the characters around it, while substituted and deleted characters are not necessarily correlated with any of the other characters.
>
> Regarding the second question, in most cases the distributions of substitution and deletion errors are skewed which can lead to unrealistic biases when generating synthetic data (with some errors being critically underrepresented). However, we did not observe similar issues with the distributions of insertion errors.
>
> We will fix the typos you mentioned, thank you.

---

### Official Review · Reviewer_UNKv · 2023-08-04

**Soundness:** 4

**Excitement:**

3: Ambivalent: It has merits (e.g., it reports state-of-the-art results, the idea is nice), but there are key weaknesses (e.g., it describes incremental work), and it can significantly benefit from another round of revision. However, I won't object to accepting it if my co-reviewers champion it.

**Paper Topic And Main Contributions:**

The paper is proposing a novel approach to generating synthetic data, which is required for training a transcription-error correcting model. In particular, the authors a) demonstrate effectiveness of modelling the synthetic data closer to the observed error distribution as compared to standard randomised approaches; and b) show that adjusting BPE vocabulary is central to the language-specific data synthesis for transcribed error correction. The paper’s main contributions are computationally-aided linguistic analysis and new synthetic data resources.

**Reasons To Accept:**

The field of ASR error correction is strongly limited by the absence of high quality training data, which would reproduce the fine error patterns, but simultaneously would allow large models to generalise well for the unseen cases. The novel technique for distribution-matched and BPE-adjusted data synthesis is shown to be effective for the task of transcribed error correction and is of high interest for the community. The authors’ argumentation, analysis and experimental set-up appear reliable and useful for the challenge they are addressing.

**Reasons To Reject:**

The effectiveness of approach is demonstrated solely on a subset (mostly European) languages. This limits the generalisibility of the results.

**Reproducibility:**

2: Would be hard pressed to reproduce the results. The contribution depends on data that are simply not available outside the author's institution or consortium; not enough details are provided.

**Reviewer Confidence:**

3: Pretty sure, but there's a chance I missed something. Although I have a good feel for this area in general, I did not carefully check the paper's details, e.g., the math, experimental design, or novelty.

---

> ### Author Rebuttal · Authors · 2023-08-28
>
> Thank you for your constructive feedback.
>
> We chose to focus on European languages for two primary reasons. Firstly, we aimed to use solely publicly available and well-known datasets to enable an easy reproduction of our results. Secondly, we have observed that our approach correlates more closely with phonetic and morphological attributes rather than the structural aspects of languages, and from this perspective the experimented languages are diverse.
>
> We believe that our findings and approaches can be applied to other languages and are not bound to specific language structures.
>
> Also, our method can be easily reproduced, as we rely entirely on publicly available tools (e.g., SentencePiece package) and fully describe the implementation details.

---

### Official Review · Reviewer_AQUj · 2023-08-06

**Soundness:** 4

**Excitement:**

4: Strong: This paper deepens the understanding of some phenomenon or lowers the barriers to an existing research direction.

**Paper Topic And Main Contributions:**

- A technique to simulate error in automatic speech recognition system, which aims to mimic the error distribution of an ASR model and is tailored to specific languages.
- The synthetic data generated from the proposed technique could be used to train error correction models with notable performance improvements. Surprisingly, the error distribution learnt from one ASR model could improve on another ASR model transcription.


**Questions For The Authors:**

1. Regarding to the transferability of error distribution obtained from XLS-R model to Whisper, could you provide more detail analyses to support? Such as in terms of model architecture, training objective, training data, or some statistics on the errors made by these models, maybe they are highly overlapped, etc. And could this transferability also potentially applied to other ASR models?

**Reasons To Accept:**

- A very well written paper with effective method and comprehensive ablation studies across popular languages, ASR models.
- The use of open-source data is also a plus point.

**Reasons To Reject:**

- The fact that the error distribution learnt from one ASR model could improve the performance on another ASR model is very interesting, however, it is not thoroughly analyzed. The authors hypothesized that the approach could "capture unique patterns specific to speech recognition". This is a strong claim.

**Reproducibility:**

4: Could mostly reproduce the results, but there may be some variation because of sample variance or minor variations in their interpretation of the protocol or method.

**Reviewer Confidence:**

4: Quite sure. I tried to check the important points carefully. It's unlikely, though conceivable, that I missed something that should affect my ratings.

**Typos Grammar Style And Presentation Improvements:**

- Line 231: frequency of c' -> frequency of c
- Line 490: with only ~25K hours -> with only ~25 hours

---

> ### Author Rebuttal · Authors · 2023-08-28
>
> Thank you for your comments and for pointing out these important issues.
>
> Indeed, we observed some degree of overlap between the error distributions of different ASR models.
>
>
>
> Upon closer examination we observed that many errors are characterized by mispronunciation by the speaker (such as merging words or omitting characters) and adverse audio conditions.
>
> These gaps are increasingly bridged as the ASR models’ capacity and training set sizes increase. That is, different models tend to fail in similar spots, yet the stronger models overcome some of the errors.
>
> These observations can explain the error distribution overlap and the ability to transfer the learned mapping of the errors to other models.
>
> We also noticed that the error distributions are more informative when extracted using medium to smaller models (in terms of model size and training data size), and that the smaller models’ distributions can assist larger models (e.g., one of the Whisper variants), but not necessarily the other way around.
>
> These observations are indeed very interesting, yet we left it outside the paper’s scope due to the page limitation. We will use the extra page which could be added upon acceptance to address these issues and provide an ablation demonstrating these findings. We will also add an appendix with a comparison of the error distributions extracted using different models and settings and fix the typos and grammar issues.

---

### Meta-Review · Area_Chair_2mXa · 2023-08-31

**Recommendation:** 5
**Confidence:** 4

**Metareview:**

**Summary:**
The paper introduces a novel approach for generating synthetic data to train error correction models for automatic speech recognition (ASR) systems. The proposed technique mimics the error distribution of ASR models, offering noticeable performance improvements in training error correction models. This approach demonstrates the ability to transfer error distributions across different ASR models. The paper presents convincing results across multiple languages, highlighting the importance of vocabulary size and token length adjustments for optimal language-specific data synthesis. Notably, the proposed approach outperforms a language model-based post-processing method for error correction.

**Pros:**

- The paper provides a clear description of the proposed technique for generating synthetic data, aligning it with ASR error distributions (Reviewer 1, Reviewer 2).



- Comprehensive ablation studies across popular languages, ASR models and publicly available well-known datasets demonstrate the effectiveness of the proposed approach (Reviewer 1, Reviewer 2).



- The approach's ability to transfer error distributions across different ASR models is intriguing and potentially of high interest to the community (Reviewer 1).



- The study covers multiple languages, showcasing the language-dependent nature of optimal tokenizer vocabulary size and token length (Reviewer 3).



- The paper offers a clear and coherent discussion of the findings and contributions (Reviewer 3).


**Cons:**

- The potential transferability of error distributions across different ASR models is intriguing but is not thoroughly analyzed, which could be seen as a limitation (Reviewer 1).



- The paper's effectiveness is demonstrated primarily on a subset of languages, raising concerns about the generalizability of results (Reviewer 2).



- Some details for reproducibility might be lacking, as indicated by one reviewer, potentially affecting the broader applicability of the approach (Reviewer 2).



- The reasons behind the contextual nature of insertion errors versus deletions and substitutions should be further elaborated (Reviewer 3).

Reviewer 1: AQUj,
Reviewer 2: UNKv,
Reviewer 3: z7J7

---

### Decision · Program_Chairs · 2023-10-07

**Decision:**

Accept-Main

**Comment:**

**Summary:**
The paper introduces a novel approach for generating synthetic data to train error correction models for automatic speech recognition (ASR) systems. The proposed technique mimics the error distribution of ASR models, offering noticeable performance improvements in training error correction models. This approach demonstrates the ability to transfer error distributions across different ASR models. The paper presents convincing results across multiple languages, highlighting the importance of vocabulary size and token length adjustments for optimal language-specific data synthesis. Notably, the proposed approach outperforms a language model-based post-processing method for error correction.

**Pros:**

- The paper provides a clear description of the proposed technique for generating synthetic data, aligning it with ASR error distributions (Reviewer 1, Reviewer 2).



- Comprehensive ablation studies across popular languages, ASR models and publicly available well-known datasets demonstrate the effectiveness of the proposed approach (Reviewer 1, Reviewer 2).



- The approach's ability to transfer error distributions across different ASR models is intriguing and potentially of high interest to the community (Reviewer 1).



- The study covers multiple languages, showcasing the language-dependent nature of optimal tokenizer vocabulary size and token length (Reviewer 3).



- The paper offers a clear and coherent discussion of the findings and contributions (Reviewer 3).


**Cons:**

- The potential transferability of error distributions across different ASR models is intriguing but is not thoroughly analyzed, which could be seen as a limitation (Reviewer 1).



- The paper's effectiveness is demonstrated primarily on a subset of languages, raising concerns about the generalizability of results (Reviewer 2).



- Some details for reproducibility might be lacking, as indicated by one reviewer, potentially affecting the broader applicability of the approach (Reviewer 2).



- The reasons behind the contextual nature of insertion errors versus deletions and substitutions should be further elaborated (Reviewer 3).

Reviewer 1: AQUj,
Reviewer 2: UNKv,
Reviewer 3: z7J7